# HealthSLM-Bench: Benchmarking Small Language Models for On-device Healthcare Monitoring

## Abstract

On-device healthcare monitoring play a vital role in facilitating timely interventions, managing chronic health conditions, and ultimately improving individuals' quality of life. Previous studies on large language models (LLMs) have highlighted their impressive generalization abilities and effectiveness in healthcare prediction tasks. However, most LLM-based healthcare solutions are cloud-based, which raises significant privacy concerns and results in increased memory usage and latency. To address these challenges, there is growing interest in compact models, Small Language Models (SLMs), which are lightweight and designed to run locally and efficiently on mobile and wearable devices. Nevertheless, how well these models perform in healthcare prediction remains largely unexplored. We systematically evaluated SLMs on health prediction tasks using zero-shot, few-shot, and instruction fine-tuning approaches, and deployed the best performing fine-tuned SLMs on mobile devices to evaluate their real-world efficiency and predictive performance in practical healthcare scenarios. Our results show that SLMs can achieve performance comparable to LLMs while offering substantial gains in efficiency, reaching up to $17\times$ lower latency and $16\times$ faster inference speed on mobile platforms. However, challenges remain, particularly in handling class imbalance and few-shot scenarios. These findings highlight SLMs, though imperfect in their current form, as a promising solution for next-generation, privacy-preserving healthcare monitoring. Our code is available at https://anonymous.4open.science/r/health-SLM-C1B0/.

## 1 Introduction

The proliferation of mobile and wearable devices, coupled with recent advances in deep learning, has significantly advanced the landscape of continuous health monitoring (Dinh-Le et al., 2019; Pham et al., 2022; Jia et al., 2024; Wu et al., 2023). These technologies enable a range of real-time applications, from the detection of physiological anomalies (Gabrielli et al., 2025) to the delivery of personalized interventions (Ghadi et al., 2025). Meanwhile, large language models (LLMs) have demonstrated remarkable generalization in processing heterogeneous data and performing diverse downstream tasks (Ferrara, 2024; Imran et al., 2024). Early studies indicate that LLM-based analysis can provide a deeper contextual interpretation of sensor data and enable more adaptive health monitoring systems compared to conventional approaches (Khasentino et al., 2025).

Despite this promise, major obstacles impede the practical deployment of LLM-driven wearable health solutions. Current approaches usually depend on cloud-based inference, necessitating data transmission to external servers, which raises concerns around user privacy, data security, and communication latency (Das, 2025; Li et al., 2024; Xu et al., 2025; Wang et al., 2025). Alternatively, on-device deployment is hindered by severe resource constraints typical of mobile and wearable hardware, as well as the real-time requirements of health applications, rendering full-sized LLMs infeasible for timely inference. These challenges highlight a critical need for efficient, privacy-preserving techniques that achieve competitive performance with LLMs, while being suitable for deployment on resource-limited mobile and wearable devices.

Small Language Models (SLMs) present a promising alternative by reducing memory consumption and facilitating deployment on mobile and wearable devices. On-device inference with SLMs not only lowers communication latency but also enhances the protection of sensitive personal data, while maintaining competitive performance on natural language processing tasks (Microsoft, 2024; Zhang et al., 2024a; Qwen, 2024; Team & DeepMind, 2024). Nevertheless, their ability to interpret sensor data from mobile and wearable devices and accurately infer health conditions in real-world settings remains an open question. Although prior work (Wang et al., 2024) has demonstrated the feasibility of using SLMs on mobile devices to predict simple health status (e.g., fatigue, sleep quality), there is still a lack of comprehensive benchmarking that thoroughly evaluates SLMs for a wide range of health applications.

To bridge this gap, we present a comprehensive benchmark, HealthSLM-Bench, which aims to evaluate a variety of state-of-the-art (SOTA) SLMs on a suit of health prediction tasks spanning four publicly available datasets. Our benchmark systematically assesses model performance using three evaluation protocols: zero-shot, few-shot, and instruction-based fine-tuning. To assess practical feasibility, we further deploy top-performing fine-tuned models on mobile devices and rigorously evaluate their on-device efficiency in terms of memory usage and inference latency. Experimental results demonstrate that SLMs can achieve comparable performance compared with SOTA healthcare LLMs across ten healthcare monitoring tasks, while substantially reducing memory and latency overheads. Our main contributions are as follows:

- We introduce, HealthSLM-Bench, an extensive benchmark that systematically evaluates nine SOTA SLMs on ten health prediction tasks across four real-world mobile and wearable datasets.

- We investigate various evaluation paradigms, including zero-shot, few-shot, and instruction-based fine-tuning, providing a comprehensive performance analysis under different adaptation scenarios.

- We demonstrate the feasibility of deploying fine-tuned SLMs on resource-constrained mobile devices and quantify their efficiency in terms of real-world memory and latency footprints.

## 2 RELATED WORK

**LLMs for health monitoring.** With the rise of mobile and wearable devices, a variety of human-centered sensing signals can be continuously collected, enabling ongoing monitoring of human health in daily life. Recent studies have shown that the physical status data collected by mobile devices is strongly associated with health status (Ballinger et al., 2018; Hallgrímsson et al., 2019; Mullick et al., 2022). Their work demonstrates how passive wearable sensor data can be effectively utilized to predict depression in adolescents using traditional ML models. However, these approaches, typically trained on specific datasets or tailored architectures, often struggle to generalize across heterogeneous tasks, and contexts (Kasl et al., 2024). LLMs, powered by their generalization capabilities, have shown great success in the healthcare domain. For example, Health-LLM (Kim et al., 2024) and MultiEEG-GPT (Hu et al., 2024b) demonstrate the effectiveness of leveraging LLMs in healthcare monitoring through textual and physiological data. Instead of just deploying these models directly for healthcare applications, recent work has explored domain adaptation strategies such as few-shot prompting, instruction tuning, and domain-specific fine-tuning to improve performance on medical tasks (Xu et al., 2024). Notably, PaLM2 (Singhal et al., 2023) illustrates the benefits of combining diverse adaptation strategies (e.g. few-shot and fine-tuned) across medical datasets. Meanwhile, evaluations of GPT-4 highlight that SOTA LLMs may reduce the reliance on extensive adaptation, as they already demonstrate strong capacity for medical reasoning with limited supervision (Nori et al., 2023). More recently, applied systems such as PhysioLLM (Fang et al., 2024) have integrated LLMs with wearable sensor data to provide personalized health insights, highlighting their adaptability across users and contexts. However, despite these advances, their computational overhead makes them impractical for privacy-sensitive, real-time mobile healthcare monitoring.

**Small Language Models.** SLMs are defined as models that are smaller in scale relative to the widely recognised LLMs, typically comprising no more than 7 billion parameters (Hu et al., 2024a). Recent research has highlighted the efficiency and strong task performance of SLMs as lightweight

Table 1: An Example of Prompt Construction for Zero-shot learning. $Z_S$ represents "Zero-shot".

| Context | Prompt |
|---|---|
| **Instruction** | You are a personalized healthcare agent trained to predict fatigue which ranges from 1 to 5 based on physiological data and user information. |
| **Main Query** | The recent 14-days sensor readings show: {14} days sensor readings show: Steps: {"1476.0, 4809.0, ..., NaN"} steps, Burned Calories: {"169.0, 419.0 ..., NaN"} calories, Resting Heart Rate: {"53.24, 52.24, ..., 51.40"} beats/min, Sleep Minutes: {"110.0, 524.0, ..., 481.0"} minutes, [Mood]: 3 out of 5. What would be the predicted fatigue level? |
| **Output Constraints** | The predicted fatigue level is: |

$$Prompt\ Z_S = Instruction_{Z_S} + \underline{Main\ query} + Output\ Constraints \qquad (1)$$

alternatives to LLMs, particularly for deployment in resource-constrained environments (Lu et al., 2025; Murthy et al., 2023). For example, Phi-3-mini-4k-Instruct, developed by Microsoft (Microsoft, 2024), contains 3.8 billion parameters and is trained on a curated blend of synthetic and high-quality public datasets, emphasizing reasoning capabilities. TinyLlama-1.1B (TinyLlama, 2024) builds on Llama 2 through parameter reduction and subsequent fine-tuning using UltraChat, a broad synthetic dialogue dataset. Similarly, Google's Gemma2-2B (Google, 2024), based on Gemini research, demonstrates robust results in text generation, summarization, and reasoning benchmarks. SmolLM-1.7B from HuggingFace (HuggingFaceTB, 2024) further diversifies training by leveraging synthetic educational materials and a breadth of domain samples, and Qwen2-1.5B (Qwen, 2024) achieves SOTA performance in both coding and mathematics despite its small footprint. Meta's Llama-3.2 series (Meta AI, 2024) continues this trend by releasing 1B and 3B parameter models designed for edge applications. While these developments affirm the viability of SLMs for a range of natural language processing tasks, the current literature leaves the open question of how effectively these compact models generalize to health prediction tasks. This is especially salient for high-stakes applications in healthcare, where accuracy and timeliness are paramount.

In comparison, our study addresses this gap by conducting comprehensive evaluations of SLMs on mobile platforms, using detailed efficiency metrics to assess their practical feasibility for mobile health monitoring applications across various datasets, model structures, and tasks.

## 3 HEALTHSLM-BENCH

We benchmark a variety of SLMs for mobile and wearable health applications using zero-shot and few-shot learning which enables in-context learning with a limited number of task-specific examples. Additionally, we instruction-tune these models on health datasets, aiming to significantly enhance their effectiveness for healthcare monitoring tasks.

### 3.1 ZERO-SHOT AND FEW-SHOT LEARNING

**Zero-shot learning.** In the zero-shot learning setting, models were evaluated without prior exposure to any example inputs during inference. Each model was provided only with a task instruction, a main query describing the 14-day summary of sensor readings, and explicit output constraints (e.g., restricting output labels for fatigue to values within the range [1–5]), as shown in Table 1. This setup was designed to evaluate the intrinsic ability of the models to interpret and respond to healthcare-related queries based solely on task instructions. The zero-shot protocol thus serves as a baseline for performance, providing a reference point for subsequent experiments involving few-shot learning and instruction tuning.

**Few-shot learning.** Few-shot learning (Brown et al., 2020) was employed to enhance task comprehension by augmenting the model inputs with a small set of labeled examples. Unlike zero-shot

Table 2: An Example of Prompt Construction for Few-shot learning. $Z_S$ and $F_S$ represent "Zero-shot" and "Few-shot", respectively.

| Context | Prompt |
|---|---|
| **Instruction** | You are a health assistant. Your mission is to read the following examples and return your prediction based on the health query. |
| **Examples** | $<$example 1$>$, $<$example 2$>$, ... $<$example $N>$ |
| **Question** | Finally, please answer to the below question: $<Prompt\ Z_S>$ |

$$Examples = (Prompt\ Z_S + Answer)_N \qquad (2)$$
$$Prompt\ F_S = Instruction_{F_S} + Examples + Prompt\ Z_S \qquad (3)$$

Table 3: Summary of the four health wearable sensor datasets used in our experiments.

| Dataset | Participants | Duration | Collection Methods | Derived Tasks | Task Types |
|---|---|---|---|---|---|
| PMData | 16 | 5 months | Fitbit Versa 2 | Fatigue, Readiness, Stress, Sleep Quality | Classification |
| LifeSnaps | 71 | 4 months | Fitbit Sense + EMA | Stress Resilience, Sleep Disorder | Regression / Classification |
| GLOBEM | 497 | 3 years | Fitness tracker + mobile app | Depression, Anxiety | Classification |
| AW_FB | 46 | 104 hours | GENEActiv, Apple Watch S2, Fitbit HR2 | Calories, Activity | Regression / Classification |

learning, which relies solely on the model's generalized knowledge, this approach leverages in-context learning to better interpret task-specific data. As shown in Table 2, the few-shot prompt ($Prompt\ F_S$), formalized in Equation 3, consists of an explicit instruction $Instruction_{F_S}$, a set of $N$ example pairs ($Prompt\ Z_S + Answer)_N$, and the target query $Prompt\ Z_S$. Specifically, the $Instruction_{F_S}$ directs the model to review the $N$ examples before responding to the target query. Each example follows the same structure as the zero-shot prompt, i.e., consisting of a task instruction and a main query, but also includes the corresponding answer. This design enables the model to ground its predictions in observed input–output patterns, capturing relationships that may be less apparent in a zero-shot setting. In our experiments, we varied the number of examples $N \in \{1, 3, 5, 10\}$ to examine its impact on performance, aiming to identify the most effective configuration. To maximize on-device efficiency, we did not implement chain-of-thought reasoning (CoT) (Wei et al., 2022b) and self-consistency (SC) (Wang et al., 2022), as both introduce additional token generation and computational overhead that limit practicality on resource-constrained edge devices.

## 3.2 Instructional Tuning

Instructional tuning adapts language models to follow task-specific instructions by further training them on curated instruction–response pairs (Wei et al., 2022a). Unlike zero-shot or few-shot learning, which relies on a sole task description or in-context prompts at inference time, instructional tuning updates the model parameters themselves, enabling more robust and persistent task alignment. Specifically, the instruction–response pairs were formatted using the Alpaca-style template (Taori et al., 2023), which provides a lightweight and standardized structure widely adopted in instruction-tuning benchmarks (Kim et al., 2024; Wang et al., 2023; Conover et al., 2023; Team, 2023). To enable efficient fine-tuning, we employed Low-Rank Adaptation (LoRA) (Hu et al., 2022), which introduces trainable low-rank decomposition matrices into the attention and feed-forward layers while keeping the original weights frozen. LoRA is particularly well-suited for on-device inference, as it allows effective model adaptation with minimal memory and computational overhead.

## 4 Experimental Setup

### 4.1 Datasets

We evaluate our methods using four health wearable sensor datasets: PMData (Thambawita et al., 2020), LifeSnaps (Yfantidou et al., 2022), GLOBEM (Xu et al., 2023), and AW-FB (Fuller, 2020). These datasets were collected with devices including Fitbit Versa2(Fitbit Inc., 2019), Fitbit Sense (Fitbit, Inc., 2020), GENEActiv (Activinsights Ltd., 2015), Apple WatchSeries2 (Apple Inc., 2016), and Fitbit Charge HR2 (Fitbit Inc., 2016), monitored over a study-specific duration.

Each dataset integrates wearable-derived features (e.g., steps, calories burned, resting heart rate, sleep metrics) with self-reported health status such as fatigue, stress, and readiness. From these, we derived a total of ten tasks, compromising both classification and regression, that reflect real-world health monitoring scenarios, as summarized in Table 3. For health event prediction, we format the temporal sequences of these features into 14-day windows and incorporate them into query prompts to generate predictions. The predictions produced by SLMs are then compared with the self-reported ground-truth labels. Additional dataset details, task definitions, and label distributions are provided in the Appendix.

## 4.2 MODELS

We selected nine SOTA SLMs ranging from 1B to 4B parameters, including Google's Gemma-2-2B-it (Google, 2024), Microsoft's Phi-3-mini-4k-instruct and Phi-3.5-mini (Microsoft Corporation, 2024), HuggingFace's SmolLM-1.7B (HuggingFaceTB, 2024), Alibaba's Qwen2-1.5B and Qwen2.5-1.5B (Qwen, 2024), TinyLlama's TinyLlama-1.1B (Team, 2024), and Meta-Llama's Llama-3.2-1B and Llama-3B (Meta AI, 2024). Detailed information about each dataset and SLM is provided in the Appendix.

## 4.3 IMPLEMENTATION DETAILS

**Data processing.** Following previous work (Kim et al., 2024; Wang et al., 2024; Jia et al., 2025), we standardize all datasets into daily sequences spanning 14-day windows. Task-specific labels are assigned accordingly. Each dataset is extracted, randomly shuffled, and split into training and testing subsets in an 8:2 ratio. The tasks are categorized as either classification (fatigue, readiness, sleep quality, stress, anxiety, depression, activity) or regression (calories). The label distributions for each task are provided in the Appendix.

**Model deployment.** To assess efficiency and feasibility, we deploy the top-performing health-domain–adapted SLMs, which is adapted for the health domain and instructional tuned using health-related datasets, on an iPhone 15 Pro Max equipped with 8 GB of RAM. These models are converted to the GGUF format (Generalized Graphical Unified Format) (Face, 2023) to ensure compatibility with lightweight inference engines such as Llama.cpp (Ggerganov). Due to the strict memory constraints of mobile devices, we apply 4-bit quantization to enable efficient deployment. As shown in prior studies (Murthy et al., 2023), quantization lowers computational costs while maintaining most of the model's performance. Both the conversion and quantization steps are performed using Llama.cpp (Gerganov & community, 2023).

**Evaluation metrics.** To evaluate model performance under *zero-shot*, *few-shot*, and *instructional-tuning* settings, we use mean absolute error (MAE) for regression tasks and accuracy for classification tasks. For efficiency evaluation of mobile deployment, we assess the models latency using metrics such as Time-to-First-Token (TTFT), Input Tokens Per Second (ITPS), Output Tokens Per Second (OTPS), and Output Evaluation Time (OET) and Total Time. In addition, We also track CPU and RAM usage to evaluate on-device resource consumption. Further details are provided in the Appendix.

## 5 RESULTS AND DISCUSSION

We compare the performance of SLMs and SOTA LLMs under the same settings as in (Kim et al., 2024).

## 5.1 OVERALL PERFORMANCE

**Zero-shot learning.** As shown in Table 4, SLMs achieve comparable or better performance than LLMs across the four health datasets. For stress prediction, SLMs achieve a lower mean MAE of 0.61, compared to 0.64 for LLMs, where lower values indicate better performance. SLMs also outperform LLMs in readiness and fatigue prediction, with a mean MAE of 2.15 for SLMs versus 2.56 for LLMs, and a higher mean accuracy of 52.2% for SLMs compared to 41.54% for LLMs. For other tasks, including stress resilience, sleep disorder, sleep quality, anxiety, depression, and activity,

Table 4: Performance of LLMs and SLMs under **zero-shot (ZS)** across ten healthcare monitoring tasks. **STRS**: Stress, **READ**: Readiness, **FATG**: Fatigue, **SQ**: Sleep Quality, **SR**: Stress Resilience, **SD**: Sleep Disorder, **ANX**: Anxiety, **DEP**: Depression, **ACT**: Activity, **CAL**: Calorie Burn. Best is **bold**, second-best is underlined. '-' denotes failure to produce a valid prediction.

| | | PMData | | | | LifeSnaps | | GLOBEM | | AW-FB | |
| --- | --- | --- | --- | --- | --- | --- | --- | --- | --- | --- | --- |
| | **Model** | **STRS** (↓) | **READ** (↓) | **FATG** (↑) | **SQ** (↓) | **SR** (↓) | **SD** (↑) | **ANX** (↓) | **DEP** (↓) | **ACT** (↑) | **CAL** (↓) |
| **LLMs** | MedAlpaca | 0.76 | 2.18 | 46.8 | 0.68 | 1.17 | 40.3 | 1.23 | 0.89 | 21.7 | 35.0 |
| | PMC-Llama | 1.33 | 4.83 | 0.0 | 2.25 | 1.21 | 41.7 | 2.33 | 2.23 | - | 43.4 |
| | Asclepius | 0.43 | **1.44** | 27.3 | 0.45 | - | - | **0.82** | 1.10 | - | **28.9** |
| | ClinicalCamel | 0.40 | 2.11 | 58.1 | **0.37** | 1.35 | 88.3 | 0.97 | 0.79 | 16.3 | 43.4 |
| | Flan-T5 | **0.36** | 1.82 | 56.8 | 0.56 | 2.20 | 57.1 | 2.84 | 2.89 | 23.4 | 66.0 |
| | Palmyra-Med | 0.83 | 5.01 | 43.5 | 0.44 | 1.03 | 3.13 | 2.07 | 1.99 | **29.7** | 75.3 |
| | Llama 2 | 0.57 | 2.86 | 41.2 | 0.89 | - | - | 1.19 | 1.23 | - | - |
| | BioMedGPT | 0.37 | 2.12 | 61.2 | 0.41 | **0.77** | - | 0.95 | **0.85** | 12.2 | - |
| | BioMistral | 0.55 | 2.12 | 56.6 | 0.45 | 1.59 | **90.0** | 0.90 | - | 18.4 | 41.0 |
| | GPT-3.5 | - | 2.38 | 70.8 | 0.87 | 1.21 | 19.0 | - | - | 13.8 | 36.4 |
| | GPT-4 | - | 2.22 | **72.2** | 0.73 | 1.23 | - | - | - | 22.6 | 75.2 |
| | Gemini-Pro | 0.79 | 1.69 | 34.0 | 0.78 | 2.67 | 84.6 | 1.03 | 0.95 | 17.7 | 31.4 |
| | Mean | 0.64 | 2.56 | 41.5 | 0.60 | 1.44 | 53.0 | 1.43 | 1.44 | 19.5 | 47.6 |
| **SLMs** | gemma-2-2b-it | 0.72 | 2.07 | 52.8 | 0.47 | 1.59 | - | 0.91 | **0.53** | - | 105.12 |
| | Phi-3-mini-4k | 0.45 | **1.52** | 62.9 | 0.48 | 1.28 | **80.0** | 1.08 | 1.26 | 17.4 | 93.80 |
| | SmolLM-1.7B | 1.42 | 2.99 | 11.0 | 1.00 | - | 44.4 | 2.59 | 2.87 | **21.7** | 277.21 |
| | Qwen2-1.5B | **0.39** | 2.03 | 63.2 | **0.45** | 2.29 | 55.6 | 1.42 | 1.65 | 14.1 | 185.22 |
| | TinyLlama-1.1B | 0.43 | 2.06 | 51.2 | 0.47 | - | 44.4 | 2.40 | 2.58 | 19.7 | 198.72 |
| | Llama-3.2-1B | 0.40 | 1.87 | **63.8** | 0.69 | **1.25** | 42.2 | 1.51 | 1.85 | 11.7 | 280.32 |
| | Llama-3.2-3B | 0.67 | 2.24 | 40.8 | 0.46 | 1.63 | 44.4 | 1.26 | 0.75 | 15.7 | **19.7** |
| | Phi-3.5-mini | 0.40 | 2.34 | 61.2 | **0.45** | 1.41 | 73.3 | **0.88** | 0.84 | 15.4 | 56.8 |
| | Qwen2.5-1.5B | 0.56 | 2.25 | 62.9 | 0.93 | 2.12 | 55.6 | 1.36 | 1.63 | 15.7 | 72.20 |
| | Mean | 0.61 | 2.15 | 52.2 | 0.60 | 1.65 | 55.0 | 1.49 | 1.55 | 16.4 | 143.23 |

SLMs perform within a similar range to LLMs. Among the SLMs, Gemma-2-2B-it and Phi-3-mini-4k consistently deliver strong results for fatigue and readiness, while Qwen2.5-1.5B matches or exceeds LLM performance on several tasks. However, SLMs do have some limitations. SmolLM-1.7B often underperforms relative to LLMs, and most SLMs struggle with calorie estimation, where the mean MAE is 143.23 for SLMs compared to 47.6 for LLMs, suggesting that regression tasks may be more challenging for SLMs.

*In sum, under zero-shot settings, SLMs generally match or surpass LLMs on most health prediction tasks, notably achieving better results in stress, readiness, and fatigue predictions. Leading SLMs, such as Gemma-2-2B-it and Phi-3-mini-4k, show consistent strength compared with SOTA LLMs.*

**Few-shot learning.** The few-shot (FS) results are shown in Table 5. For LLMs, we compare the best few-shot performance (FS-best) to SLMs using a range of few-shot sample counts (1, 3, 5, 10) in SLMs. As shown in Table 5, even when provided with in-context examples in the one-shot setting (FS-1), SLMs demonstrate competitive performance compared to their larger counterparts across multiple healthcare monitoring tasks, and also outperforms zero-shot SLMs on average.

Comparing the performance across different few-shot settings reveals interesting patterns in SLM behavior. In the FS-1 setting, SLMs achieve competitive performance levels compared to LLMs across most tasks. For instance, SLMs achieve a mean of 0.47 for stress prediction compared to LLMs' 0.90, and 0.49 for sleep quality compared to LLMs' 0.72. As the number of few-shot examples increases from FS-1 to three-shot (FS-3), five-shot (FS-5), and ten-shot (FS-10), the performance shows task-dependent variations. For stress prediction, the mean performance remains relatively stable across all few-shot settings. Similarly, sleep quality prediction maintains consistent performance throughout the different few-shot configurations.

However, certain tasks exhibit different response patterns to increased few-shot examples. Anxiety and depression prediction tasks show notable improvement as the number of examples increases from FS-1 to FS-3, with further refinement observed in subsequent settings. This suggests that mental health prediction tasks may benefit more from additional contextual examples compared to physiological monitoring tasks when using SLMs without fine-tuning, which has also been observed in recent work comparing SLMs and LLMs in mental health prediction tasks (Jia et al., 2025). As shown in Table 5, we also observed that the collapse pattern appears at FS-1, FS-3, and FS-5, but does not occur at FS-10. This phenomenon was observed only in PMData and LifeSnaps tasks,

Table 5: Performance of LLMs and SLMs under **few-shot (FS)** setting across across ten healthcare monitoring tasks. **STRS**: Stress, **READ**: Readiness, **FATG**: Fatigue, **SQ**: Sleep Quality, **SR**: Stress Resilience, **SD**: Sleep Disorder, **ANX**: Anxiety, **DEP**: Depression, **ACT**: Activity, **CAL**: Calorie Burn. Best result is in **bold**, second-best result is underlined. '-' denotes model failed to produce valid prediction.

| | Model | PMData | | | | LifeSnaps | | GLOBEM | | AW-FB | |
| --- | --- | --- | --- | --- | --- | --- | --- | --- | --- | --- | --- |
| | | STRS (↓) | READ (↓) | FATG (↑) | SQ (↓) | SR (↓) | SD (↑) | ANX (↓) | DEP (↓) | ACT (↑) | CAL (↓) |
| **LLMs** (FS-best) | MedAlpaca | 0.78 | 1.94 | 36.2 | 0.69 | 0.94 | 49.6 | **0.97** | **0.56** | 19.3 | 36.7 |
| | GPT-3.5 | 0.94 | **1.62** | 73.9 | 0.77 | 0.80 | 58.5 | 1.98 | 0.68 | **26.3** | 26.5 |
| | GPT-4 | **0.76** | 1.64 | 61.3 | **0.60** | **0.45** | 73.4 | 1.11 | 0.60 | 15.4 | **24.0** |
| | Gemini-Pro | 1.10 | 2.20 | 24.8 | 0.80 | 1.18 | 71.8 | 1.30 | 1.05 | 15.0 | 37.2 |
| | Mean | 0.90 | 1.85 | 49.1 | 0.72 | 0.84 | 63.3 | 1.34 | 0.72 | 19.0 | 31.1 |
| **SLMs** (FS-1) | gemma-2-2b-it | **0.41** | 2.30 | **59.9** | **0.45** | 1.72 | 55.6 | 2.04 | 2.40 | 0.0 | 24.22 |
| | Phi-3-mini-4k | 0.43 | 1.56 | 47.8 | 0.46 | **0.61** | 62.2 | 1.99 | **1.94** | 21.4 | 21.58 |
| | SmolLM-1.7B | **0.41** | 1.31 | 51.5 | 0.46 | 0.66 | 55.6 | 3.12 | 3.46 | **22.1** | 19.94 |
| | Qwen2-1.5B | **0.41** | 1.29 | 51.5 | 0.46 | 0.65 | 44.4 | 2.15 | 2.47 | 14.4 | 19.07 |
| | TinyLlama-1.1B | **0.41** | 1.30 | 51.5 | 0.46 | 0.66 | 44.4 | 3.10 | 3.39 | 14.0 | 18.97 |
| | Llama-3.2-1B | 0.55 | 1.50 | 51.5 | 0.65 | 0.65 | 44.4 | 2.32 | 3.03 | 20.4 | 18.43 |
| | Llama-3.2-3B | 0.79 | 1.87 | 28.8 | 0.54 | 1.28 | 71.1 | **1.84** | 2.01 | 18.1 | 37.45 |
| | Phi-3.5-mini | **0.41** | 1.36 | 51.5 | 0.46 | 1.04 | **91.1** | 3.06 | 3.42 | 14.4 | 51.33 |
| | Qwen2.5-1.5B | 0.43 | **1.28** | 54.5 | 0.47 | 1.26 | 71.1 | 3.10 | 3.44 | 14.7 | **18.04** |
| | Mean | 0.47 | 1.53 | 49.8 | 0.49 | 0.95 | 60.0 | 2.52 | 2.84 | 15.5 | 25.45 |
| **SLMs** (FS-3) | gemma-2-2b-it | 0.48 | 1.66 | 44.8 | 0.49 | 1.65 | 53.3 | - | - | - | - |
| | Phi-3-mini-4k | 0.41 | 1.67 | 44.8 | **0.45** | 0.57 | 75.6 | **0.88** | 0.54 | 19.4 | 55.0 |
| | SmolLM-1.7B | - | - | - | - | 0.74 | 53.3 | **0.87** | 0.58 | 15.4 | 19.0 |
| | Qwen2-1.5B | 0.41 | 1.68 | **51.5** | 0.46 | 0.66 | 57.8 | 0.88 | **0.54** | 23.1 | 19.8 |
| | TinyLlama-1.1B | - | - | - | - | 0.82 | 44.4 | 2.93 | 3.04 | 14.4 | 17.9 |
| | Llama-3.2-1B | 0.43 | 1.73 | 49.8 | 0.54 | 1.53 | 44.4 | 0.88 | 0.54 | 15.4 | 18.5 |
| | Llama-3.2-3B | 0.41 | 1.78 | **51.5** | 0.47 | 1.01 | 42.2 | 1.19 | 1.12 | **24.1** | 19.3 |
| | Phi-3.5-mini | 0.41 | 1.42 | **51.5** | 0.46 | 1.02 | **84.4** | 0.91 | 0.55 | **24.1** | 32.9 |
| | Qwen2.5-1.5B | **0.39** | 1.44 | 37.1 | 0.76 | 0.72 | 55.6 | 1.36 | 0.64 | 18.1 | **17.9** |
| | Mean | 0.42 | 1.62 | 47.3 | 0.52 | 0.97 | 56.8 | 1.24 | 0.94 | 19.2 | 25.0 |
| **SLMs** (FS-5) | gemma-2-2b-it | 0.48 | 1.35 | 61.5 | 0.47 | 1.75 | 53.3 | - | - | - | - |
| | Phi-3-mini-4k | 0.41 | **1.32** | 57.2 | 0.49 | **0.70** | 66.7 | 0.88 | 0.56 | 22.1 | 37.3 |
| | SmolLM-1.7B | - | - | - | - | 0.78 | 44.4 | **0.87** | 0.76 | 17.1 | **18.6** |
| | Qwen2-1.5B | 0.41 | 1.41 | 51.5 | **0.46** | 0.83 | 55.6 | 1.20 | 1.12 | 20.4 | 29.4 |
| | TinyLlama-1.1B | - | - | - | - | 1.14 | 42.2 | 3.15 | 3.51 | **24.1** | 37.0 |
| | Llama-3.2-1B | 0.43 | 1.42 | 52.5 | **0.46** | 1.62 | 42.2 | 1.18 | 1.38 | 15.1 | 27.2 |
| | Llama-3.2-3B | 0.41 | 1.59 | 52.2 | **0.46** | 1.14 | 46.7 | 1.18 | 1.23 | 18.4 | 28.5 |
| | Phi-3.5-mini | 0.41 | 1.41 | 51.5 | **0.46** | 1.00 | **68.9** | 1.46 | 1.56 | **24.1** | 23.7 |
| | Qwen2.5-1.5B | **0.39** | 1.44 | 41.5 | 0.49 | 0.93 | 57.8 | 1.28 | 1.52 | 17.4 | 28.5 |
| | Mean | 0.42 | 1.42 | 52.6 | 0.47 | 1.10 | 53.1 | 1.40 | 1.45 | 19.8 | 28.8 |
| **SLMs** (FS-10) | gemma-2-2b-it | 0.49 | **1.40** | **63.6** | 0.50 | 0.75 | 64.4 | 1.23 | 1.09 | - | - |
| | Phi-3-mini-4k | 1.01 | 1.70 | 32.8 | **0.45** | 0.51 | **71.1** | 0.82 | 0.63 | 17.7 | 18.5 |
| | SmolLM-1.7B | - | - | - | - | 0.78 | 44.4 | **0.77** | **0.53** | 15.1 | 19.1 |
| | Qwen2-1.5B | **0.41** | 1.55 | 56.2 | 0.46 | 0.65 | 55.6 | 0.87 | 0.54 | 17.7 | 18.0 |
| | TinyLlama-1.1B | - | - | - | - | - | - | - | - | 21.1 | **17.2** |
| | Llama-3.2-1B | 0.89 | 1.61 | 8.4 | 0.46 | 0.71 | 37.8 | 0.87 | 0.77 | 15.7 | 19.5 |
| | Llama-3.2-3B | 0.49 | 1.83 | 39.8 | 0.47 | **0.49** | 64.4 | 2.04 | 1.23 | 19.1 | 18.1 |
| | Phi-3.5-mini | 0.42 | **1.40** | 34.1 | 0.48 | 0.60 | 46.7 | 0.77 | 1.10 | **22.1** | 18.9 |
| | Qwen2.5-1.5B | 0.66 | 2.47 | 33.4 | 0.50 | 0.63 | 57.8 | 0.87 | 0.54 | 17.4 | 19.1 |
| | Mean | 0.62 | 1.71 | 38.3 | 0.48 | 0.68 | 55.3 | 1.03 | 0.80 | 18.2 | 18.5 |

such as stress, fatigue, sleep quality and sleep disorder, while readiness remained unaffected and no collapse was noted in GLOBEM or AW-FB tasks. Upon further inspection, this trend is likely attributed to the limited representation of labels when only a small number of few-shot examples are provided. For this reason, the collapse disappears at FS-10, as more examples enable a more representative range of labels. The label distribution for identical predicted values is provided in the Appendix.

*Overall, SLMs perform competitively with LLMs in few-shot healthcare tasks, even with just one example. More examples help models achieve more stable and reliable performance.*

**Instruction tuning.** As shown in Table 6, both SLMs and SOTA LLMs (Kim et al., 2024) are instruction-tuned, yet SLMs outperform LLMs in tasks such as fatigue and calorie estimation. Specifically, SLMs achieve higher best values for fatigue and activity, while also attaining lower estimation error for readiness and calorie burn, demonstrating their superior accuracy for these important health measures. Although LLMs perform slightly better in stress, sleep quality, stress resilience, anxiety and depression prediction, with lower mean values for Sleep quality and anxi-

Table 6: Performance of LLMs and SLMs under **instruction tuning (LoRA)** setting across ten healthcare monitoring tasks. **STRS**: Stress, **READ**: Readiness, **FATG**: Fatigue, **SQ**: Sleep Quality, **SR**: Stress Resilience, **SD**: Sleep Disorder, **ANX**: Anxiety, **DEP**: Depression, **ACT**: Activity, **CAL**: Calorie Burn. Best result is in **bold**, second-best result is underlined. '-' denotes model failed to produce valid prediction.

| | | PMData | | | LifeSnaps | | GLOBEM | | AW-FB | |
|---|---|---|---|---|---|---|---|---|---|---|
| | Model | STRS (↓) | READ (↓) | FATG (↑) | SR (↓) | SD (↑) | ANX (↓) | DEP (↓) | ACT (↑) | CAL (↓) |
| **LLMs** (lora) | HealthAlpaca-lora-7b | 0.53 | **1.40** | 50.0 | 0.58 | **0.62** | 61.7 | **0.62** | **0.51** | 27.4 | 43.6 |
| | HealthAlpaca-lora-13b | **0.34** | 1.56 | **54.8** | **0.39** | 0.70 | 92.0 | 1.04 | 0.67 | **29.0** | **39.6** |
| | Mean | 0.44 | 1.48 | 52.4 | 0.49 | 0.66 | 76.9 | 0.83 | 0.59 | 28.2 | 41.6 |
| **SLMs** (lora) | gemma-2-2b-it | - | - | - | 0.51 | 0.72 | - | 1.27 | 1.02 | **34.4** | **2.80** |
| | Phi-3-mini-4k | **0.40** | 2.14 | 62.2 | 0.52 | 0.97 | **68.9** | **0.81** | 0.71 | 22.4 | 9.67 |
| | SmolLM-1.7B | 0.93 | 1.68 | 15.4 | 0.89 | 1.49 | 44.4 | 0.84 | 0.54 | 16.1 | 18.87 |
| | Qwen2-1.5B | 0.43 | 1.52 | 62.2 | **0.47** | 0.90 | 55.6 | 0.92 | 0.97 | 18.7 | 5.21 |
| | TinyLlama-1.1B | **0.40** | 1.30 | **63.2** | **0.47** | **0.67** | 55.6 | 0.83 | 0.67 | 22.1 | 5.51 |
| | Llama-3.2-1B | 0.43 | 2.25 | 49.8 | 0.81 | 1.09 | 48.9 | 0.86 | **0.54** | 19.2 | 5.78 |
| | Llama-3.2-3B | 0.60 | 1.53 | 40.8 | **0.47** | 0.86 | 53.3 | 0.88 | **0.54** | 22.1 | 3.64 |
| | Phi-3.5-mini | 0.49 | 1.55 | 62.2 | 0.92 | 0.98 | 62.2 | 0.88 | 0.66 | 19.4 | 12.09 |
| | Qwen2.5-1.5B | 0.87 | 1.49 | 13.0 | 0.87 | 1.89 | 55.6 | 1.04 | 0.79 | 21.7 | 4.57 |
| | Mean | 0.57 | 1.68 | 46.1 | 0.66 | 1.06 | 55.6 | 0.93 | 0.72 | 21.8 | 7.57 |

Table 7: Efficiency & Utilization of LLMs and SLMs across datasets. **Mean token** denotes the average number of prompt tokens in the 10 selected samples from that dataset.

| | Model | TTFT(s) | ITPS(t/s) | OET(s) | OTPS(t/s) | Total Time(s) | CPU(%) | RAM(GB) |
|---|---|---|---|---|---|---|---|---|
| **PMData** (Mean token: 720) | Phi-3-mini-4k | 7.15 | 100.42 | 1.08 | 12.08 | 8.47 | **40.77** | 6.27 |
| | TinyLlama-1.1B | **1.37** | **527.01** | **0.35** | **45.89** | **1.79** | 44.86 | **5.51** |
| | Llama-2-7b | 24.03 | 29.99 | 4.32 | 3.84 | 28.79 | 317.70 | 7.04 |
| **LifeSnaps** (Mean token: 487) | Phi-3-mini-4k | 4.23 | 113.47 | 1.34 | 14.15 | 5.87 | **38.12** | 6.16 |
| | TinyLlama-1.1B | **0.94** | **517.87** | **0.33** | **46.93** | **1.34** | 45.27 | **5.45** |
| | Llama-2-7b | 13.68 | 35.65 | 3.73 | 5.09 | 17.87 | 262.99 | 7.03 |
| **GLOBEM** (Mean token: 236) | Phi-3-mini-4k | 1.82 | 128.75 | 1.10 | 15.66 | 3.16 | **35.60** | 6.23 |
| | TinyLlama-1.1B | **0.45** | **519.62** | **0.37** | **49.27** | **0.84** | 41.96 | **5.52** |
| | Llama-2-7b | 4.71 | 50.14 | 2.39 | 7.95 | 7.50 | 298.93 | 7.05 |
| **AW_FB** (Mean token: 152) | Phi-3-mini-4k | 1.18 | 127.70 | 1.17 | 16.19 | 2.42 | **34.21** | 6.01 |
| | TinyLlama-1.1B | **0.29** | **523.90** | **0.32** | **50.06** | **0.63** | 43.46 | **5.48** |
| | Llama-2-7b | 2.94 | 51.74 | 1.98 | 8.95 | 5.31 | 270.79 | 6.97 |

ety , these differences are relatively modest compared to the clear advantages of SLMs in fatigue and calorie estimation. For other tasks such as stress, stress resilience, depression, and depression, both SLMs and LLMs show similar performance, with only minor differences in best values. Notably, SLMs like TinyLlama-1.1B and Phi-3-mini-4k stand out for their strong and consistent results across multiple tasks. For the less-performing cases (e.g., sleep quality, anxiety and sleep disorder) of SLMs, we observed that SLMs tend to predict only the majority classes without attempting to predict the minority classes (i.e., class-imbalance bias; *cf*. Appendix), causing the model to stuck at sub-optimal performance on those tasks.

*In sum, these findings demonstrate that SLMs, when properly tuned, are not only competitive but often superior to LLMs for specific healthcare tasks, particularly fatigue and calorie estimation. This highlights the potential of SLMs for efficient, accurate, and large-scale healthcare applications, making them a compelling choice where resource efficiency and task-specific performance are essential.*

## 5.2 DEPLOYMENT EFFICIENCY

To investigate efficiency and computational cost in real-world deployment, we ran inference with the two top-performing models, Phi-3-mini-4k and TinyLlama-1.1B, which were instructional-tuned using LoRA, on an iPhone 15 Pro Max with 8GB memory capacity. Since the SOTA LLM HealthAlpaca-lora-7b (Kim et al., 2024) did not release its checkpoint, we compared the on-device performance of selected SLMs against the baseline Llama-2-7b (the backbone of HealthAlpaca-lora-7b) using PMData to evaluate deployment efficiency. For fair comparison, we random select a total

of ten samples from the four health datasets for both Llama-2-7b, Phi-3-mini-4k and TinyLlama-1.1 to evaluate latency and hardware utilization.

**PMData.**    As shown in Table 7, the efficiency results of the two instruction-tuned SLMs on PMData (720 tokens) demonstrate that SLMs preserve their latency and memory advantages over Llama-2-7b. Both SLMs outperform Llama-2-7b in latency and throughput. Specifically, Phi-3-mini-4k achieves a $3.4\times$ faster time-to-first-token (TTFT) and a $24\times$ faster output evaluation time (OET), with gains of over $+250\%$ in both input tokens per second (ITPS) and output tokens per second (OTPS), resulting in a $3.4\times$ faster total time. TinyLlama-1.1B shows even larger margins, with $17.5\times$ faster TTFT, $12\times$ faster OET, and more than $+1600\%$ ITPS, leading to an impressive $16.1\times$ faster total time compared to Llama-2-7b. The memory footprint of the SLMs is also much smaller. Specifically, Phi-3-mini-4k uses $11\%$ less RAM, and TinyLlama-1.1B uses $22\%$ less than Llama-2-7b. Between the two SLMs, Phi-3-mini-4k offers moderate efficiency gains in some metrics but is consistently slower than TinyLlama-1.1B by about $4\times$, suggesting that efficiency is strongly tied to model size, with smaller models generally providing superior benefits.

**LifeSnaps, GLOBEM, and AW_FB.**    On LifeSnaps, GLOBEM, and AW_FB, reduced token inputs led to lower latency across all models, yet SLMs still showed clear efficiency advantages over Llama-2-7b. TinyLlama-1.1B achieves $1.46\times$, $3.04\times$, and $4.7\times$ faster TTFT on LifeSnaps (487 tokens), GLOBEM (236 tokens), and AW_FB (152 tokens), while Phi-3-mini-4k achieves $1.69\times$, $3.93\times$, and $6.1\times$, respectively. In these datasets, both SLMs substantially outperform Llama-2-7b, with TTFT up to $14.6\times$ faster. Throughput metrics such as ITPS, OET, and OTPS remain mostly consistent, indicating throughput is relatively insensitive to input length. Overall, shorter-input datasets yield lower prediction times mainly due to reduced input processing.

**Robustness to Input Length.**    Compared to SLMs, Llama-2-7b is less robust to longer inputs. In latency evaluation, it lags further behind SLMs on LifeSnaps and PMData than on GLOBEM and AW_FB. Its throughput on PMData drops to approximately $50\%$ of that on GLOBEM and AW_FB, and to $70\%$ of LifeSnaps, suggesting sensitivity to long sequences. Since throughput should remain stable across datasets, as demonstrated by SLMs, this degradation likely stems from out-of-memory pressure (Zhang et al., 2024c; Lee et al., 2024; Zhang et al., 2024b), where heavy workloads force KV-cache spills into slower system memory. By contrast, the smaller footprint of SLMs allows them to tolerate longer inputs. For hardware utilization, RAM usage remains largely unchanged ($\leq 4\%$) for both SLMs and LLMs, while CPU utilization decreases by about $10\%$ on shorter-input datasets.

*Together, these findings show that SLMs achieve substantial reductions in both input processing latency and generation latency, especially hold clear advantages on longer-context datasets. In contrast, LLMs (even at 7B) suffer substantial slowdowns under constrained RAM capacity, making SLMs an ideal and practical solution for resource-constrained mobile health applications.*

## 6    CONCLUSION AND FUTURE WORK

In this paper, we introduce HealthSLM-Bench, a comprehensive benchmark designed to systematically evaluate a range of SOTA SLMs on healthcare monitoring tasks under zero-shot, few-shot, and instruction-tuning scenarios. Furthermore, we assess the efficiency of these models following instruction-tuning through on-device deployment experiments. Our study shows that SLMs can match or even surpass much larger LLMs after adapted with few-shot and instructional tuning while delivering superior efficiency gain, making them practical for real-time on-device deployment. At the same time, we also identified their limitations in few-shot prompting and restricted effectiveness in instruction tuning, particularly under class-imbalanced datasets. Both limitations point to several promising directions for future work. One is to investigate the underlying causes of the few-shot anomaly and explore robust prompt design to prevent collapse. Another is to explore imbalance-aware training approaches, for example by adjusting loss weighting or augmenting minority-class samples, to reduce class bias during SLM fine-tuning. Additionally, leveraging adaptive techniques such as test-time adaptation could further strengthen SLM generalisation in health applications. Overall, our benchmark establishes SLMs as a promising yet imperfect solution for efficient and privacy-preserving healthcare applications, motivating further exploration to address these challenges.

ETHICS STATEMENT

This work uses only publicly available wearable-sensor datasets collected with participant consent. No personally identifiable information was accessed, stored, or released in the course of this study.

REPRODUCIBILITY STATEMENT

To ensure reproducibility, we provide detailed dataset preprocessing, hyperparameter selection, and training procedures in the Appendix. In addition, we also use the deterministic decoding strategy for SLMs generation to ensure consistent outputs and strengthen reproducibility. Our code is available at `https://anonymous.4open.science/r/health-SLM-C1B0/`. The full repository and scripts required to replicate our experiments will be released publicly upon publication, along with instructions to reproduce all reported results.

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

APPENDIX

## A USE OF LLMS

In preparing this paper, we used LLMs exclusively as a language refinement tool, particularly assisted with minor text polishing, including improving grammar, sentence flow, and word choice. They were not used for research ideation, experiment design, implementation, data analysis, or substantive writing. All technical contributions, experiments, and results presented in this paper are entirely the work of the authors.

## B IMPLEMENTATION DETAILS

We fine-tune our SLMs on a NVIDIA A100 80GB GPUs with a batch size of 128 with 3 number of epochs for the purpose of fine-tuning, with Adam optimizer and a learning rate as 5e-5 (cosine learning rate scheduler and dynamic warmup steps of 5% of dataset size). It took about 7 hours for 9 SLMs in 3 epochs of training with the default training setting. We adopt greedy decoding method with sampling set to False. We utilize the same prompt of zero-shot for LoRA tuned SLMs inference. To ensure re-productiveness, we employ the greedy decoding strategy to make the output prediction deterministic. While most language models default to sampling-based decoding (e.g., top-$k$, top-$p$), we explicitly disabled these strategies to maintain reproducibility across runs. To better simulate edge-device conditions, where computational resources are constrained, we capped the maximum number of generated tokens at 30. Generation stops once this limit is reached, even if the answer is incomplete, which balances efficiency and response quality. The codes and fine-tuned models will be made publicly available upon the release of the camera-ready version of this paper.

## C ADDITIONAL EXPERIMENTS

Table 8: Performance of LLMs and SLMs under **instruction tuning (LoRA)** and **full-parameters tuning (FT)** across ten healthcare monitoring tasks. **STRS**: Stress, **READ**: Readiness, **FATG**: Fatigue, **SQ**: Sleep Quality, **SR**: Stress Resilience, **SD**: Sleep Disorder, **ANX**: Anxiety, **DEP**: Depression, **ACT**: Activity, **CAL**: Calorie Burn. Best result is in **bold**, second-best result is underlined. '-' denotes model failed to produce valid prediction.

| | | PMData | | | | LifeSnaps | | GLOBEM | | AW-FB | |
|---|---|---|---|---|---|---|---|---|---|---|---|
| | Model | STRS (↓) | READ (↓) | FATG (↑) | SQ (↓) | SR (↓) | SD (↑) | ANX (↓) | DEP (↓) | ACT (↑) | CAL (↓) |
| **LLMs** (lora) | HealthAlpaca-lora-7b | 0.53 | **1.40** | 50.0 | 0.58 | **0.62** | 61.7 | **0.62** | **0.51** | 27.4 | 43.6 |
| | HealthAlpaca-lora-13b | **0.34** | 1.56 | **54.8** | **0.39** | 0.70 | **92.0** | 1.04 | 0.67 | **29.0** | **39.6** |
| | Mean | 0.44 | 1.48 | 52.4 | 0.49 | 0.66 | 76.9 | 0.83 | 0.59 | 28.2 | 41.6 |
| **LLMs** (FT) | HealthAlpaca-7b | 0.31 | 1.32 | **70.7** | 0.35 | 0.62 | 72.1 | **0.46** | 0.49 | 41.7 | 31.5 |
| | HealthAlpaca-13b | **0.21** | **1.08** | 61.2 | **0.14** | **0.32** | **93.9** | 0.95 | **0.24** | **51.0** | **28.5** |
| | Mean | 0.26 | 1.20 | 65.9 | 0.25 | 0.47 | 83.0 | 0.71 | 0.37 | 46.4 | 30.0 |
| **SLMs** (lora) | gemma-2-2b-it | - | - | - | 0.511 | 0.723 | - | 1.271 | 1.023 | **34.4** | **2.8** |
| | Phi-3-mini-4k | 0.398 | 2.144 | 62.2 | 0.522 | 0.966 | **68.9** | **0.809** | 0.712 | 22.4 | 9.7 |
| | SmolLM-1.7B | 0.930 | 1.676 | 15.4 | 0.893 | 1.489 | 44.4 | 0.843 | 0.539 | 16.1 | 18.9 |
| | Qwen2-1.5B | 0.428 | 1.522 | 62.2 | 0.472 | 0.903 | 55.6 | 0.923 | 0.967 | 18.7 | 5.2 |
| | TinyLlama-1.1B | **0.395** | **1.304** | 63.2 | 0.472 | **0.667** | 55.6 | 0.833 | 0.669 | 22.1 | 5.5 |
| | Llama-3.2-1B | 0.428 | 2.251 | 49.8 | 0.809 | 1.090 | 48.9 | 0.860 | 0.535 | 19.2 | 5.8 |
| | Llama-3.2-3B | 0.595 | 1.532 | 40.8 | **0.465** | 0.858 | 53.3 | 0.883 | 0.535 | 22.1 | 3.6 |
| | Phi-3.5-mini | 0.485 | 1.548 | 62.2 | 0.920 | 0.981 | 62.2 | 0.880 | 0.659 | 19.4 | 12.1 |
| | Qwen2.5-1.5B | 0.866 | 1.485 | 13.0 | 0.866 | 1.891 | 55.6 | 1.037 | 0.793 | 21.7 | 4.6 |
| | Mean | 0.57 | 1.68 | 46.1 | 0.66 | 1.06 | 55.6 | 0.93 | 0.72 | 21.8 | 7.6 |
| **SLMs** (FT) | gemma-2-2b-it | **0.351** | **1.304** | 62.9 | **0.452** | 0.625 | 55.6 | 0.883 | 0.535 | 53.2 | 2.1 |
| | Phi-3-mini-4k | 0.398 | 1.535 | 62.9 | 0.468 | 0.549 | **86.7** | **0.803** | 0.542 | 19.4 | 28.2 |
| | SmolLM-1.7B | 0.732 | 2.993 | 18.1 | 1.672 | - | 43.2 | 1.997 | 0.686 | 16.0 | - |
| | Qwen2-1.5B | 0.395 | **1.304** | 63.2 | 0.478 | 0.820 | 55.6 | 1.050 | 0.535 | 38.1 | 3.8 |
| | TinyLlama-1.1B | 0.398 | 1.331 | 63.2 | 0.455 | 0.675 | 44.4 | 0.863 | 0.535 | 38.1 | 2.3 |
| | Llama-3.2-1B | 0.395 | 2.160 | **64.5** | 0.462 | 0.681 | 55.6 | - | - | 38.5 | 2.1 |
| | Llama-3.2-3B | 0.385 | **1.304** | 56.9 | **0.452** | 0.574 | - | 0.873 | **0.532** | **55.4** | **1.7** |
| | Phi-3.5-mini | 0.535 | 1.512 | 61.2 | 0.508 | **0.450** | 80.0 | 0.886 | 0.549 | 16.4 | 6.2 |
| | Qwen2.5-1.5B | 0.682 | 1.446 | 39.6 | 0.455 | 1.394 | 71.1 | 0.953 | 0.695 | 15.4 | 5.8 |
| | Mean | 0.47 | 1.66 | 54.8 | 0.60 | 0.72 | 61.5 | 1.04 | 0.58 | 32.3 | 6.5 |

In addition to LoRA, we also conducted experiment on full-parameters tuning. As shown in the Table 8, all tasks demonstrated improvement in FT compare to LoRA. Notably, the best accuracy on activity and sleep disorder showed the significant improvements, rising from 34.4 to 55.4 and from 68.9 to 86.7, respectively, enabling our fine-tuned SLMs to surpass HealthAlpaca-7b (55.4 *vs.* 41.7 and 86.7 *vs.* 72.1) and even outperform the 13B version on activity (55.4 *vs.* 51.0). This behavior also observed on MAE tasks, such as stress resilience, where the lowest error decreased from 0.667 to 0.450, significantly outperforming HealthAlpaca-7b (0.62). calorie burn persist its advantages observed under LoRA and further bring its best error from 2.8 to 1.7, which outperform HealthAlpaca at both 7b (1.7 *vs. 31.5*) and 13b (1.7 *vs.* 28.5). For tasks on PMData and GLOBEM, the best results showed limited improvement, remaining stuck at suboptimal levels due to class-imbalance in model's predictions. However, they are still comparable to HealthAlpaca-7b in most of cases.

## D  TASK CATEGORIZATION AND LABEL DISTRIBUTION

**PMData** is a dataset that integrates life-logging and activity-logging information, comprising personalized health monitoring data collected from 16 participants over a period of five months. Using the Fitbit Versa 2 smartwatch wristband (Fitbit Inc., 2019), objective signals such as calories burned, resting heart rate, step count, sleep duration, and more were gathered. In addition, participants provided self-reported measurements of their health status via the PMSys sports logging application, such as fatigue, mood, stress, etc. In our setting, these self-reports were categorized into prediction tasks with labels for fatigue, readiness, sleep quality, and stress (Kim et al., 2024; Wang et al., 2024).

- Stress (STRS): Estimation of an individual's stress level based on physiological data and self-reported measures. (0-5, Classification)

- Readiness (READ): Assessment of an individual's readiness for physical activity/exercise. (0-10, Classification)

- Fatigue (FATG): Monitoring of signs of tiredness or exhaustion based on sports and life-log data in the last 14 days. (1-5, Classification)

- Sleep Quality (SQ): Estimation of an individual's sleep quality. (1-5, Classification)

All tasks is assessed with factors including total sleep time, Steps, mood and other sports data like Burned Calories and Resting Heart Rate over a continuous 14-day period. In terms of range, most tasks are evaluated on a scale of 1-5 or 0-5. A score of 3 represents a normal condition, and 1-2 are scores below normal states, and 4-5 are scores above normal states. For the task of readiness, the scale ranges from 0 to 10, where 0 reflects no readiness for physical activity, and 10 indicates high preparation for exercise.

The **label distribution** for each task in this dataset is shown as Figure 1.

### D.1  LIFESNAPS

LifeSnaps is a multi-modal, longitudinal, and geographically-distributed dataset designed for self-tracking physical and mental health monitoring. As stated by author (**?**), it was collected unobtrusively over a period of 4 months from 71 participants using Fitbit Sense smartwatch, validated surveys, and real-time ecological momentary assessments. The integrated Fitbit sensor data (sleep, heart rate, stress, etc), along with Ecological Momentary Assessments (context and mood, step goal, etc) enables real-time mental and physical health analysis. In this study, this dataset derives the following tasks:

- Stress Resilience (SR): Evaluation of an individual's ability to effectively cope with, positively adapt to, and recover from stress. (0.2–5, Regression)

- Sleep Disorder (SD): Identification of sleep-related irregularities in given physiological sleep patterns. (0 or 1, Classification)

Both of them are assessed using features extracted from daily wearable sensor streams over a continuous 14-day period. Specifically, the following features are used to evaluate Stress Resilience (SR):

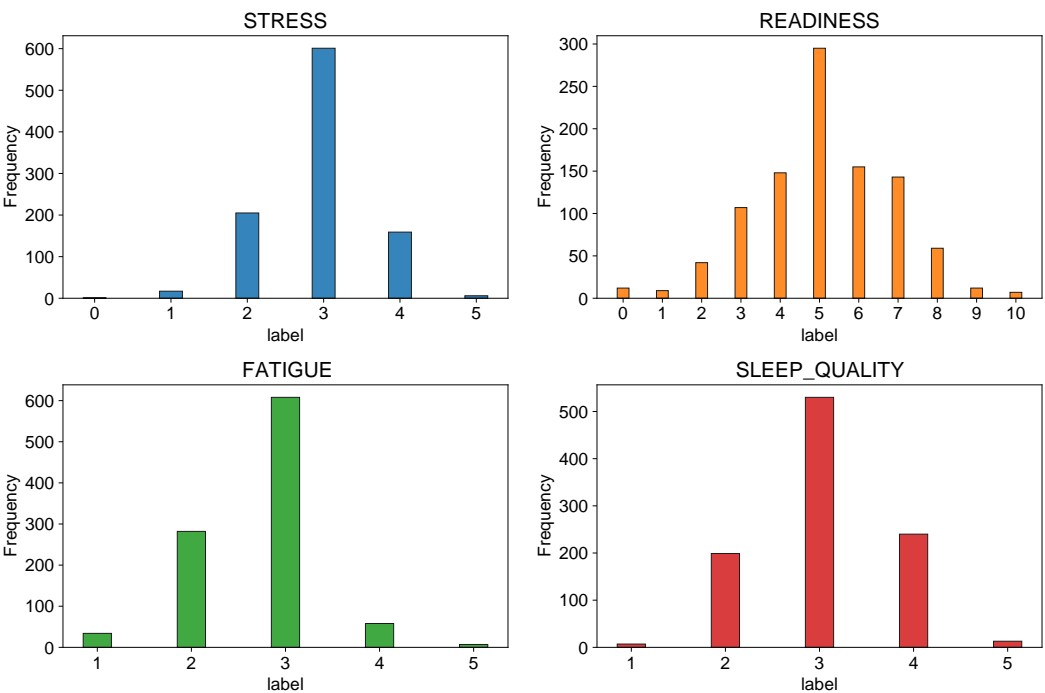

Figure 1: The label distribution of the four tasks in PMData

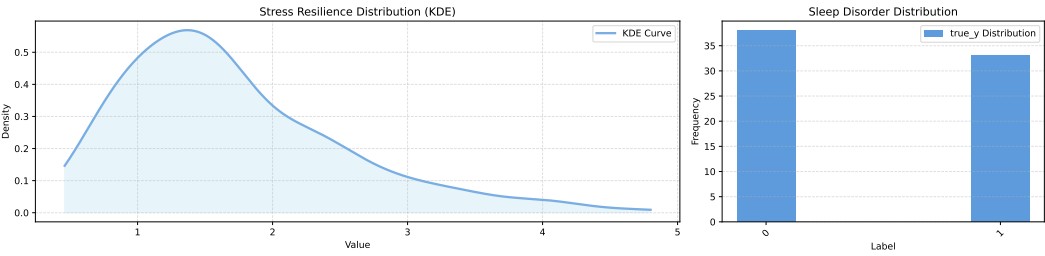

Figure 2: The Data distribution of the two tasks in LifeSnaps

Stress Score, Positive Affect Score, Negative Affect Score, Lightly Active Minutes, Moderately Active Minutes, Very Active Minutes, Sleep Efficiency, Sleep Deep Ratio, Sleep Light Ratio, and Sleep REM Ratio; For Sleep Disorder (SD), the features include: Sleep Duration, Minutes Awake, Sleep Efficiency, Sleep Deep Ratio, Sleep Wake Ratio, Sleep Light Ratio, Sleep REM Ratio, RMSSD, SpO$_2$, Full Sleep Breathing Rate, BPM and Resting Hour. In the detail of labeling, SR is a continuous value scale from 0.2 to 5, where 3.1 denotes a neutral state, values below 3.1 indicate lower resilience, and values above 3.1 suggest higher resilience to stress. SD is a binary (0,1) value, where 0 indicates the absence of disorder, and 1 denotes its presence.

The **data distribution** for each task in this dataset is shown at Figure 2.

**GLOBEM** is a passive sensing dataset for health-domain analysis. Data were gathered from 497 participants between 2018 and 2021 using a custom mobile application alongside continuous fitness tracker monitoring (24/7). This dataset captures a wide range of daily human routines, including step counts, sleep efficiency, time spent in bed after waking, time to fall asleep, and wake periods while in bed. These signals reveal associations between everyday behaviors and well-being outcomes. In our experiment, we use these behavioral signals as inputs and predict mental health conditions such as depression and anxiety (Kim et al., 2024).

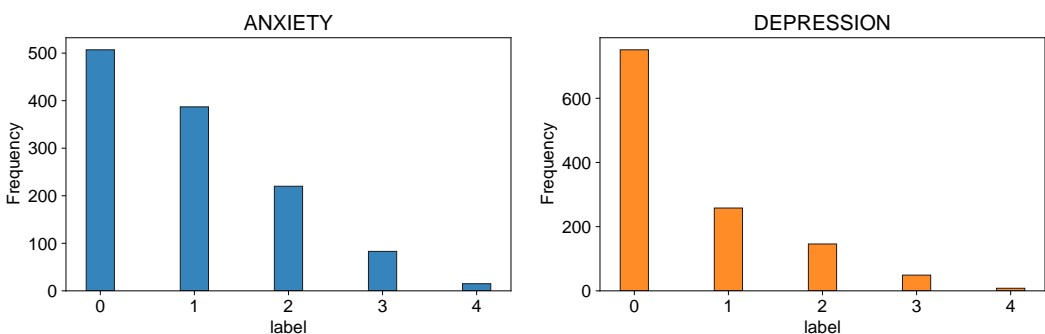

Figure 3: The label distribution of the two tasks in GLOBEM

- Depression (DEP): estimation of a depression score that analyzes patterns in user's sleeping behavior and activity levels. (0–4, Classification)

- Anxiety (ANX): estimation of an anxiety score that relies on behavioral markers such as irregular sleep patterns or heightened physiological responses, e.g. increased heart rate, reduced activity levels, and increased sleep disturbances (0–4, Classification)

Both the two tasks are assessed on the average of daily steps, sleep efficiency, duration the user stayed in bed after waking up, duration the user spent to sleep, duration the user stayed awake but still in bed, and duration the user spent to fall asleep in the last 14 days. A value of 0 implies the disorder is not present, while a value of 4 indicates severe disorder. Any values between 0 and 4 denote their severity accordingly, such as a value of 1 indicates mild disorder, 2 refers to moderate, and 3 refers to Moderately Severe.

The **label distribution** for each task in this dataset is shown at Figure 3.

**AW_FB** is a wearable dataset designed by Harvard University to study the relationship between physical activity patterns and physiological metrics, gathered from 46 participants that wear GE-NEActiv (Activinsights Ltd., 2015), Apple Watch Series 2 (Apple Inc., 2016) and a Fitbit Charge HR2 (Fitbit Inc., 2016) in a lab-based protocol. The recorded sensor data includes daily step count, heart rate, activity duration, burned calories, and metabolic equivalent of task (MET) Value. This dataset was tested to predict 6 different physical activity intensities, including lying, sitting, walking self-paced, 3 METS, 5 METS, and 7 METS.

- Activity (ACT): estimation of individual's activity intensity type based on sensor data. (0-5, Classification)

- Calories (CAL): estimation of burned calories that are expended by an individual during physical activities. (no constraint, Regression)

Activity is predicted by Steps, Burned Calories, and Heart Rate obtained during an activity period. This label ranges from 0 to 5, corresponding to Self Pace Walk, Sitting, Lying, Running 7 METs, Running 5 METs, and Running 3 METs respectively. Calories are calculated based on Steps, Heart Rate, Duration, Activity Type, and MET Value, where a higher value indicates greater energy expenditure.

The **label distribution** for each task in this dataset is shown at Figure

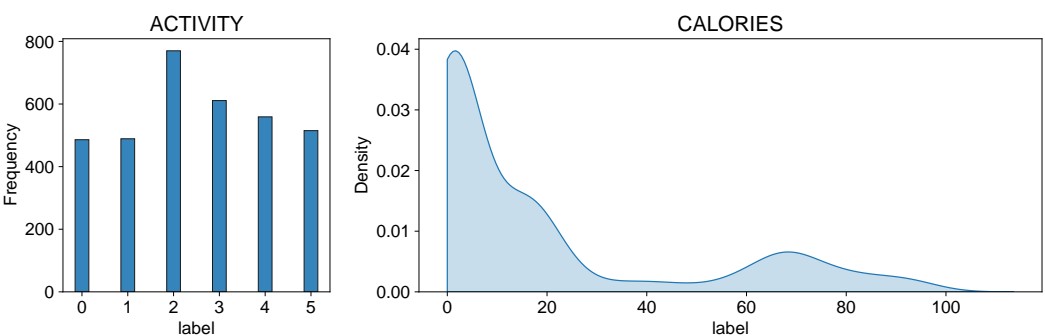

The label distribution of the two tasks in AW_FB

## E SAMPLES FROM TRAINING DATASETS

Tables 9–12 show samples derived from each of the four datasets.

Table 9: Sample instruction–response pair from the fatigue task in **PMData**.

---

**Instruction:**
You are a personalized healthcare agent trained to predict fatigue which ranges from 1 to 5 based on physiological data and user information.

**Input:**
The recent {14} days sensor readings show: Steps: {"1476.0, 4809.0, ..., NaN"} steps, Burned Calories: {"169.0, 419.0 ..., NaN"} calories, Resting Heart Rate: {"53.24, 52.24, ..., 51.40"} beats/min, Sleep Minutes: {"110.0, 524.0, ..., 481.0"} minutes, [Mood]: 3 out of 5. What would be the predicted fatigue level?

**Response:**
The predicted fatigue level is 3. `<EOS>`

---

Table 10: Sample instruction–response pair from the stress resilience task in **LifeSnaps**.

---

**Instruction:**
You are a personalized healthcare agent trained to predict stress resilience which ranges from 0.2 to 5 based on physiological data and user information.

**Input:**
The recent {7} days sensor readings show: Stress Score: {"61.0, 64.0, ..., 77.0} out of 100, [Positive Affect Score]: 39 out of 50, [Negative Affect Score]: 27 out of 50, Lightly Active Minutes: {"96.0, 126.0, ..., 173.0} minutes, Moderately Active Minutes: {"10.0, 4.0, ..., 60.0}minutes, Very Active Minutes: {"10.0, 12.0, ..., 88.0} minutes, Sleep Efficiency: {"87.0, 90.0, ..., 91.0}, Sleep Deep Ratio: {"[0.90361, 1.26667, ..., 1.25974}, Sleep Light Ratio: {"1.30932, 0.62783, ..., 0.78027}, Sleep REM Ratio: {"0.90426, 0.97647, ..., 1.31461}; What would be the predicted stress resilience index?

**Response:**
The predicted stress resilience index is 1.44. `<EOS>`

---

## F SMALL LANGUAGE MODELS

We selected 9 most state-of-the-art SLMs between 1 to 4B from top-tier tech companies. The details of each SLMs are listed below:

- **Phi-3-mini-4k-Instruct** (Microsoft, 2024): Microsoft's smallest model in the Phi-3 family. It has 3.8 billion parameters, trained on a combination of synthetic data and selected publicly available website data, with an emphasis on high-quality and reasoning-dense properties.

Table 11: Sample instruction–response pair from the Anxiety task in **GLOBEM**.

---

**Instruction:**
You are a personalized healthcare agent trained to predict PHQ-4 anxiety which ranges from 0 to 4 based on physiological data and user information.

**Input:**
The recent {14} days sensor readings show: [Steps] is 10635.9230769231. [Sleep] efficiency, duration the user stayed in bed after waking up, duration the user spent to sleep, duration the user stayed awake but still in bed, duration the user spent to fall asleep are 95.4615384615385, 0.153846153846154, 429.384615384615, 20.6153846153846, 0.0 mins in average; What would be the PHQ-4 anxiety score?

**Response:**
The predicted PHQ-4 anxiety score is 4. `<EOS>`

---

Table 12: Sample instruction–response pair from the Activity task in **AW_FB**.

---

**Instruction:**
You are a personalized healthcare agent trained to predict the type of activity among 0:"Self Pace Walk", 1:"Sitting", 2:"Lying", 3:"Running 7 METs", 4:"Running 5 METs", 5:"Running 3 METs" based on physiological data and user information.

**Input:**
The recent sensor readings show: [Steps]: 742.72 steps, [Burned Calories]: 16.46 calories, [Heart Rate]: 64.00 beats/min; What would be the predicted activity type?

**Response:**
The predicted activity type is 1. `<EOS>`

---

- **Phi-3.5-mini-Instruct** (Microsoft, 2024): A upgrade version of phi-3-mini-4k-instruct. It is built in the same architecture and dataset upon phi-3, but trained with a focus on reasoning dense data for better instruction alignment and multi-step reasoning.

- **TinyLlama-1.1B-Chat-v1.0** (TinyLlama, 2024): Distilled version of Llama 2. It uses the same architecture and tokenizer as LLaMA but is compact with 1.1 billion parameters. It was fine-tuned on the UltraChat dataset (contains field-cross synthetic dialogues generated by ChatGPT), making it compatible with a wide range of tasks.

- **Gemma2-2B-it** (Google, 2024): Google's SOTA open-source model, built on the same research and technology as the Gemini models but scaled down to 2 billion parameters. It is well-suited for text generation tasks such as question answering, summarization, and reasoning.

- **SmolLM-1.7B-Instruct** (HuggingFaceTB, 2024): HuggingFace's flagship model, it has 1.7 billion parameters and is trained on SmolLM-Corpus which consists of synthetic textbooks, stories, and educational Python and web samples.

- **Qwen2-1.5B-Instruct** (Qwen, 2024): Ailibaba's state-of-the-art SLM in Qwen2 family. It has only 1.5 billion parameters and is trained on diverse instruction-followed tasks. The included coding and mathematics data for training makes it perform well in coding and quantitative reasoning tasks.

- **Qwen2.5-1.5B-Instruct** (Qwen et al., 2025): An upgraded version of Qwen2. It is built on the same dataset and architecture, but places greater emphasis on coding and mathematics tasks, making it more optimized for reasoning and math.

- **Llama-3.2-1B-Instruct** (Meta AI, 2024): Meta-llama's state-of-the-art SLM. It shares the identical architecture and pre-trained datasets upon Llama3, but is compressed to 1B parameters.

- **Llama-3.2-3B-Instruct** (Meta AI, 2024): 3B version of Llama-3.2-1B-Instruct.

# G    TRAINING LOSS

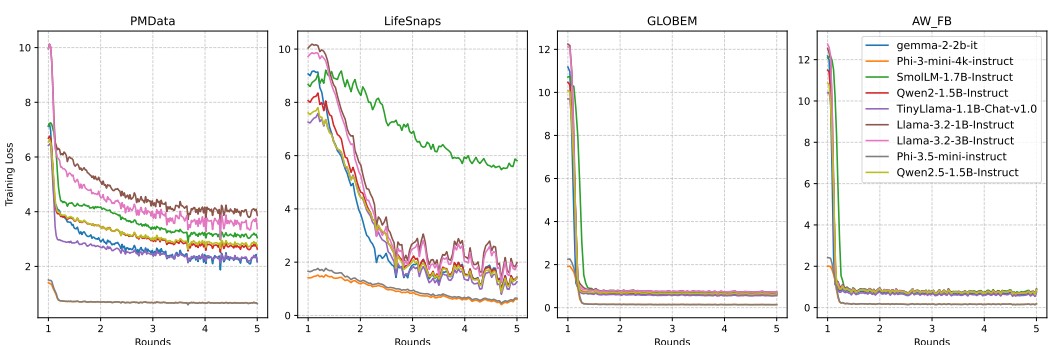

Figure 4: Training loss of 9 SLMs across 4 datasets. The training loss overall exhibits a consistent downward trend across all datasets, with larger fluctuations observed in the Lifesnaps for some SLMs.

# H    EVALUATION METRICS

## H.1    PERFORMANCE EVALUATION

For SLMs performance evaluation, Mean Absolute Error (MAE) and Accuracy are utilized to assess model prediction performance on health event prediction.

**Accuracy (%)** (Bishop, 2006) measures the proportion of correctly predicted instances out of all instances. It provides an overview of whether a model performed well overall, with higher values indicating better performance. However, accuracy does not capture the severity or magnitude of errors in misclassified cases, as all errors are treated equally.

**Mean Absolute Error (MAE)** (Hastie et al., 2009) quantifies the average magnitude of prediction errors by computing the absolute difference between predicted and actual values. Lower MAE indicates better alignment with ground truth. Unlike Accuracy, which only reflects correctness, MAE distinguishes between small and large errors. For example, predicting "3" when the true label is "4" yields an error of 1, while predicting "3" when the true label is "10" yields an error of 7. Thus, MAE captures not only whether predictions are correct but also how close incorrect predictions are to the true values.

In health event prediction, we used both Accuracy and MAE to provide complementary insights. For instance, models that achieve slightly lower accuracy but maintain consistently low MAE may be preferable, as they deliver more reliable outputs than those with higher accuracy but large error magnitudes.

**Efficiency and Utilization Evaluation** To further evaluate the efficiency and the actual latency in real cases, all state-of-the-art (SOTA) SLMs that show strong promise will be deployed in processing healthcare field data on a real iPhone 15 Pro Max. To better demonstrate the importance of efficiency on mobile devices, the widely used LLM, Llama 2, is selected and serves as a comparison to the fine-tuned SLMs.

During batch evaluations, latency metrics such as TTFT, ITPS, OTPS, OET, and Total Time are calculated as the average time spent or average token processed/generated over a sample size of $N$ (we used 10). CPU utilization is measured by the average load per second during inference, while RAM usage is reported as the maximum memory allocated to the device when the model is running.

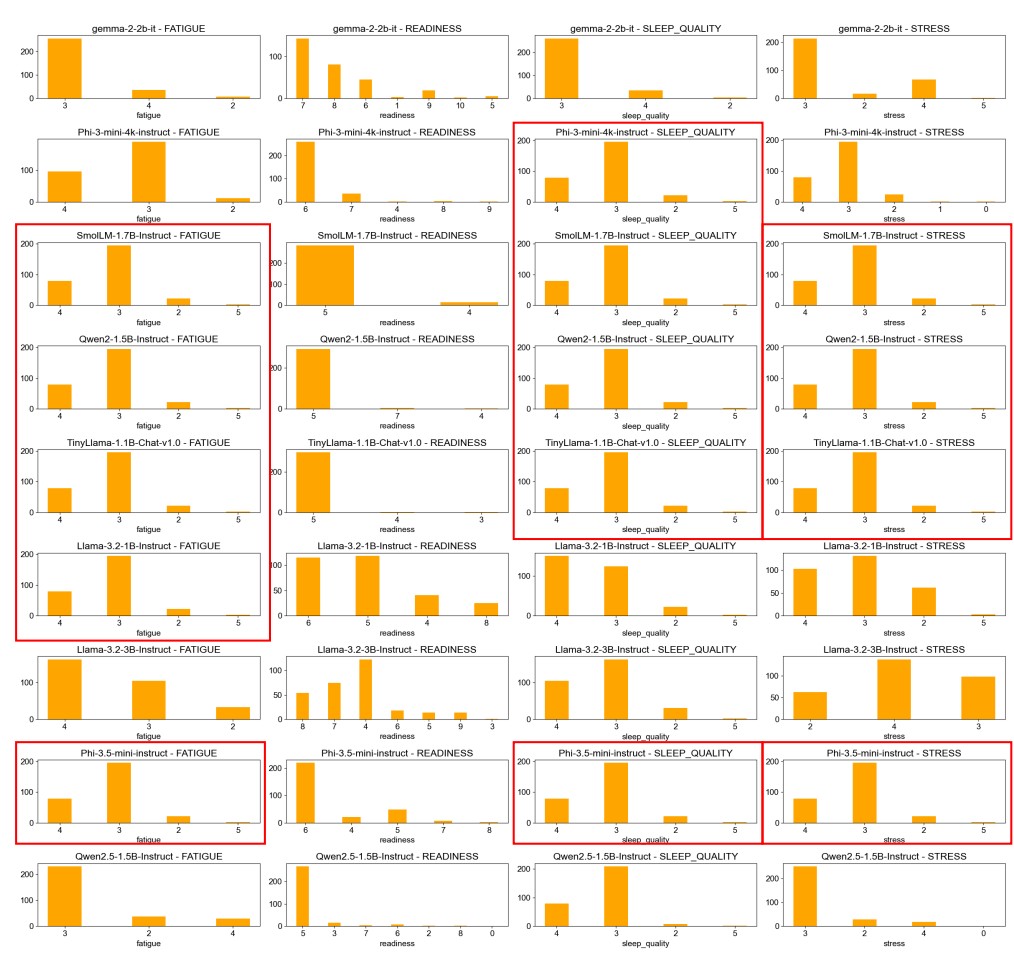

Figure 5: Distribution of predictions for the four tasks in PMData under FS setting. All collapsed predictions are highlighted in red.

# I  SLM PREDICTION DISTRIBUTION ANOMALIES

## I.1  FEW-SHOT DISTRIBUTION

The collapsed few-shot distribution are shown at Figure 5.

## I.2  INSTRUCTIONAL TUNING (LoRA) DISTRIBUTION

The instruction tuning (LoRA) prediction distribution with anomalies (class-imbalance) are shown at Figure 6 and 7.

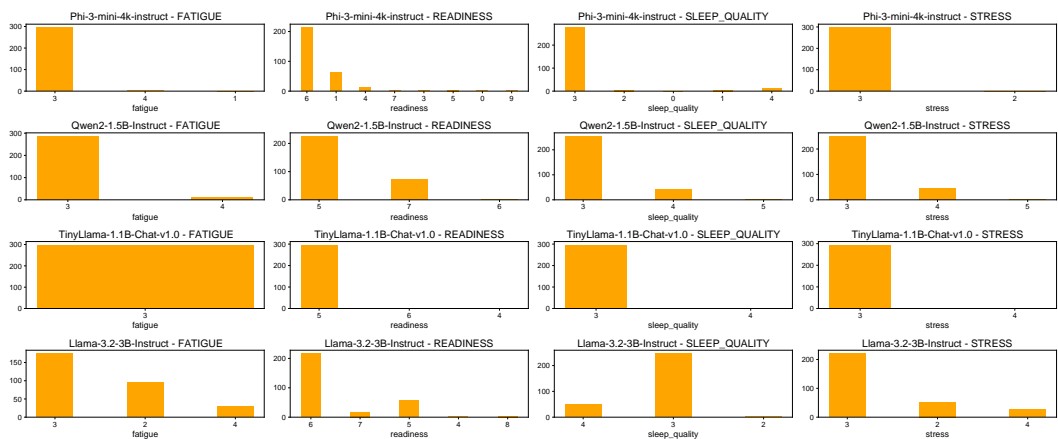

Figure 6: Distributions of model predictions across the four tasks in PMData, highlighting class imbalance.

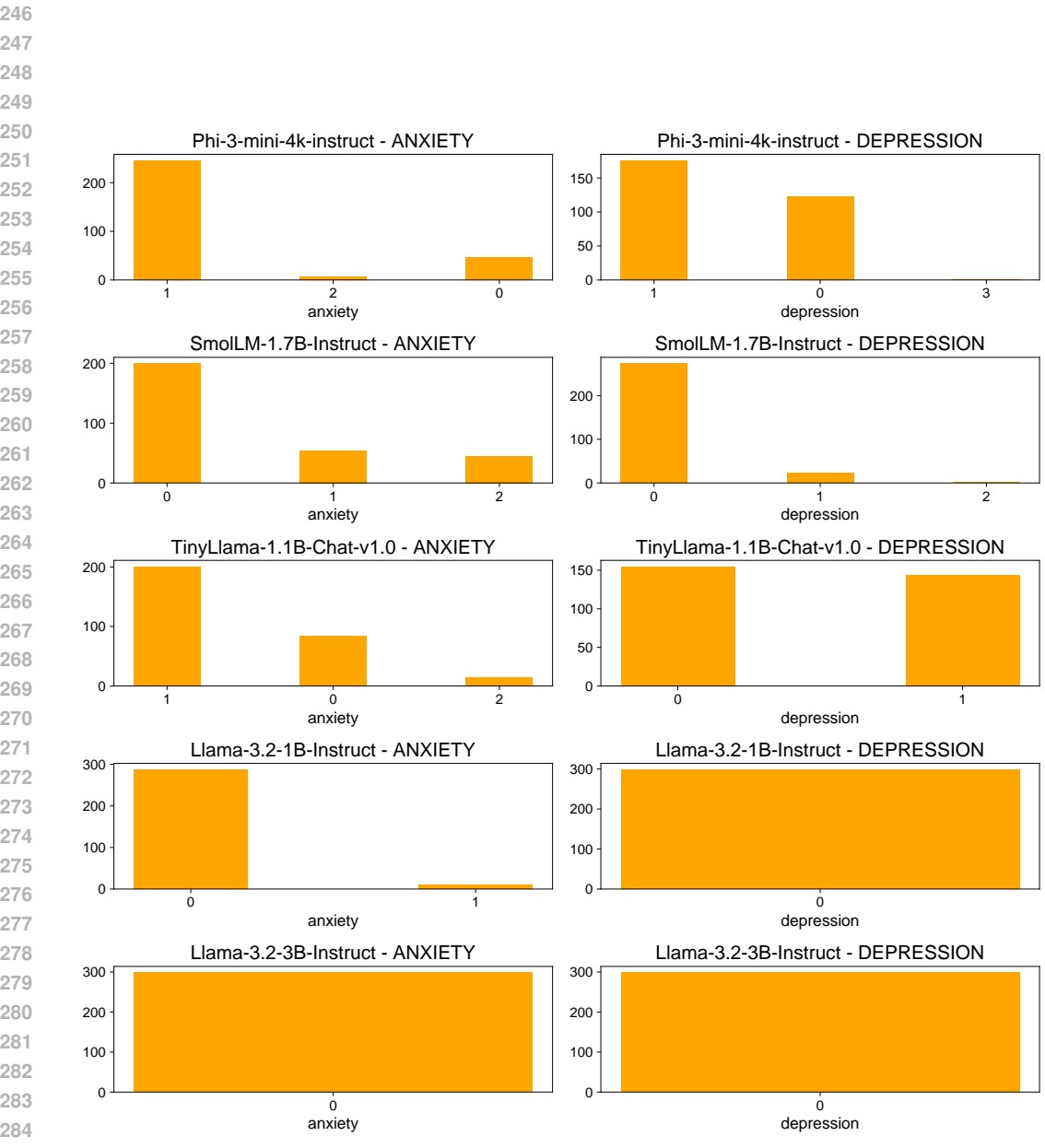

Figure 7: Distributions of model predictions across the two tasks in GLOBEM, highlighting class imbalance.

