# OpenReview forum: "HealthSLM-Bench: Benchmarking Small Language Models for On-device Healthcare Monitoring"
_ICLR.cc/2026/Conference — ICLR 2026 Conference Withdrawn Submission_

### Official Review · Reviewer_v9BL · 2025-10-30

**Soundness:** 2
**Presentation:** 2
**Contribution:** 1
**Rating:** 2
**Confidence:** 4

**Summary:**

This paper presents HealthSLM-Bench, a benchmarking framework for evaluating SLMs in mobile healthcare monitoring scenarios. The benchmark aggregates four publicly available datasets and evaluates SLM performance across zero-shot, few-shot, and instruction-tuned settings. The authors additionally deploy models on a commercial smartphone, demonstrating that SLMs can approach the performance of larger LLMs in some cases while being more efficient for on-device inference.

**Strengths:**

- The paper addresses an important and timely problem: enabling privacy-preserving and efficient inference on sensitive mobile and wearable healthcare data.
- The paper compares different LLMs and SLMs across various settings (zero-shot, few-shot, and instruction tuning).

**Weaknesses:**

- **Limited novelty in benchmark construction.** Although the paper evaluates SLMs for health monitoring, the benchmark primarily reuses four existing public datasets and follows the same 14-day windowing pipeline as prior work . Beyond adding SLM baselines, there is no new data, task design, annotation effort, or real-world sensing condition introduced. As a result, the contribution is more evaluative than a new benchmark framework.
- **Mismatch between motivation and evaluation setting.** The introduction emphasizes real-time and continuous mobile health monitoring  , but the evaluation uses 2-week aggregated summaries for inference rather than streaming or online prediction. This weakens the real-time deployment claim, as the current benchmark does not test latency-critical or streaming settings.
- **Missing comparison to relevant SOTA methods.** The paper compares general-purpose LLMs and SLMs, but does not evaluate against specialized sensor-to-language models such as SensorLM [w1] and Time2Lang [w2], which are explicitly designed to translate wearable sensor data into LLM-compatible text representations.



[w1] Zhang, Yuwei, et al. "SensorLM: Learning the Language of Wearable Sensors." arXiv preprint arXiv:2506.09108 (2025).

[w2] Pillai, Arvind, et al. "Time2Lang: Bridging Time-Series Foundation Models and Large Language Models for Health Sensing Beyond Prompting." arXiv preprint arXiv:2502.07608 (2025).

**Questions:**

- The paper reports deployment results on a single smartphone model. Can additional device types be included?
- The title uses the term *on-device*, which may imply inference directly on wearable devices (e.g., smartwatches). Since experiments run on smartphones, clarifying this-either in the title or abstract-would be less confusing.

---

### Official Review · Reviewer_W1bn · 2025-10-31

**Soundness:** 2
**Presentation:** 3
**Contribution:** 2
**Rating:** 0
**Confidence:** 4

**Summary:**

This paper proposed a medical devise small LLM benchmark to compare and test whether small LLMs can perform the required health risk prediction based on data collected from medical devices. It can monitor 10 health status based on four medical device data.

**Strengths:**

- this paper aims to address the pressing issues on how to deploy LLMs on medical device mobile app to monitor health status.
- It provides comprehensive assessment of three settings using different prompt engineering for 10 health status prediction (regression or classification).
- It develops the mobile medical device benchmark to test the performance for different LLMs, and demonstrates the feasibility of deploying small LLMs on edge devices.

**Weaknesses:**

- the major issue of this paper is the evaluation tasks, which are regression or classification tasks, then normal machine learning prediction models (such as regression models or classification models can work much better, efficient, and accurate. There is no need to involve LLMs. The authors should choose different tasks which require LLMs reasoning, and no easy existing machine learning models can be built.
- this paper did not talk about the hallucination of LLM models, especially small LLM models which have limited performance on medical reasoning.

**Questions:**

- can you please evaluate different tasks which require LLM reasonings, rather than just simple regression or classification tasks which machine learning models can do really well already.
- can you please measure the uncertainty and hallucinations for small LLMs.

---

### Official Review · Reviewer_xcdT · 2025-10-31

**Soundness:** 3
**Presentation:** 3
**Contribution:** 2
**Rating:** 2
**Confidence:** 3

**Summary:**

The paper introduces a benchmark to evaluate the performance and efficiency of Small Language Models (SLMs) for on-device healthcare monitoring. The main motivation of this paper is that traditional solutions utilizing LLMs depend on cloud-based inference, which raises issues concerning user privacy, data security, latency, and resource limitations on mobile and wearable devices. SLMs are lightweight and can be deployed on mobile devices to address the issues mentioned above. The benchmark evaluates nine state-of-the-art SLMs using zero-shot, few-shot, and instruction-based fine-tuning methods. The evaluation focuses on a total of ten health prediction tasks (including classification and regression tasks), derived from four datasets. Overall, the benchmark concludes that SLMs are a promising yet imperfect solution for an efficient and privacy-preserving healthcare monitoring solution.

**Strengths:**

1. The paper presents a benchmark for evaluating the performance of SOTA SLMs on ten health prediction tasks.
2. The evaluation is conducted under various paradigms, including zero-shot, few-shot, and instruction-based fine-tuning.

**Weaknesses:**

1. The four datasets used for evaluation are all wearable/behavior-sensing datasets and the models are evaluated mainly for classification and regression tasks. The motivation behind using LLMs/SLMs for such tasks is not clear. Moreover, it is not clear why these datasets should be considered sufficient for benchmarking the performance of SLMs for on-device healthcare monitoring?
2. The paper's contribution appears limited, as it mainly evaluates a set of LLMs and SLMs under various conditions, without proposing any new methods or concepts.

**Questions:**

1. Why the four datasets should be considered sufficient for benchmarking the performance of SLMs in on-device healthcare monitoring?
2. How does the performance of LLMs and SLMs compare to classical models in terms of performance and efficiency?

---

### Official Review · Reviewer_xcy6 · 2025-11-01

**Soundness:** 3
**Presentation:** 3
**Contribution:** 3
**Rating:** 2
**Confidence:** 4

**Summary:**

The manuscript introduces HealthSLM-Bench, a benchmark designed to evaluate small language models (SLM) in low data settings in healthcare monitoring tasks (zero-shot, few-shot,
and instruction-tuning scenarios).

Results are generated on four public health wearable sensor datasets described in Section 41. 10 tasks, consisting of classification and regression predictions, were formulated using the datasets. Each dataset is standardized into 14-day windows.

Results are evaluated using mean absolute error (MAE) for regression and accuracy for classification tasks, respectively.

The key finding of the paper is that SLMs achieve comparable performance results to LLMs, while reducing memory and latency overheads.

Ablation analyses in terms of computation efficiency and input lengths is provided in page 9. Results indicate that the two SLM models evaluated, Phi-3-mini-4k and TinyLlama-1.1B , show a significant improvement in faster time-to-first-token (TTFT) and output evaluation time (OET) metrics.

**Strengths:**

The manuscript addresses an important problem of replacing large language models (LLMs) with their smaller scale counterparts (SLMs) for the purpose of computational efficiency and wider scale access. The motivation is clearly described and manuscript covers background and related work well.

Results are evaluated across multiple public datasets, SLM and LLM models.

Code is available in an anonymized repository.

**Weaknesses:**

Authors report accuracies of 40-50% in lines 268-269, but accuracy is prevalence dependent, it is unclear what the baseline is. Accuracy may not be the best metric if the data is highly unbalanced. It may be good to additionally report random performance for comparison.

Claims in paragraph “robustness to input length” do not appear to be supported by data? Is there a result table demonstrating this ablation experiment?

The main manuscript is hard to follow, and a little difficult to connect the reported result findings to the numerical results provided. Below are some of the clarity concerns that I had.

* It is unclear which tasks/metrics are used in Table 4 (similarly for Table 5). It may be best to split by task/metric.

* For the few shot setting (lines 304-323), authors make all kinds of deductions from the results reported in Table 5, but the Table is large and very difficult to verify the conclusions. Which tasks/datasets are referred to as mental health predictions tasks? In addition, I did not understand why LLM performance is reported on FS-best while SLM performance is reported on 1,2,5, and 10 example setting. Are models compared under equitable training conditions?

* Authors should add section/subsection etc numbering for easier reading.

In summary, it is not surprising that SLMs are more computationally efficient than LLM. On the other hand, the performance comparisons do not appear to clearly demonstrate stated contributions.

**Questions:**

How does computational efficiency compare across SLM variations? Is there a difference across considered models?

What was the computer setup (memory, quantity of CPUs/GPUs/memory used to evaluate computational efficiency?

How do the results reported in Table 6 demonstrate class imbalance issues described in the conclusion/future work section? Also, which of the classes in the datasets are underrepresented/overrepresented?

---

### Note · Authors · 2025-11-12

I have read and agree with the venue's withdrawal policy on behalf of myself and my co-authors.